# Adverse childhood experiences (ACEs) associated with reduced cognitive flexibility in both college and community samples

**Vrinda Kalia** *, **Katherine Knauft, Niki Hayatbini**

Psychology Department, Miami University, Oxford, Ohio, United States of America

* kaliav@miamioh.edu

## Abstract

The prefrontal cortex is sensitive to stress experiences and significantly impacted by early life adversity. Cognitive flexibility is an executive function that is associated with positive outcomes in adulthood and implicated in activity in the prefrontal cortex. The relationship between early life adversity and cognitive flexibility is underreported. Using the cumulative risk model, we conducted two studies to examine the association between early life adversity and cognitive flexibility in college students and adults (cumulative N = 510). Exposure to early life adversity was assessed using the adverse childhood experiences scale (ACEs). Cognitive flexibility was assessed using the Wisconsin Card Sorting Test (WCST). Additionally, as perceived chronic stress is associated with impaired prefrontal cortex function, we measured that as well. Higher number of ACEs was correlated with lower number of completed categories on the WCST in both college students and adults. Perceived chronic stress was not associated with cognitive flexibility, but did correlate positively with ACEs. Individuals with a higher number of ACEs were also more likely to report higher levels of perceived chronic stress. Hierarchical regression analyses indicated that exposure to adverse childhood experiences predicted lower scores on completed categories. Our findings provide further evidence that individuals with early life adversity exhibit reduced cognitive flexibility in adulthood.

## Introduction

Unlike other primates, humans experience an extended developmental period, which is made possible by the care and protection provided by their parents [1]. Unfortunately, there are many humans who do not get their basic needs met in early development. Instead, they are exposed to early life adversity in the form of abuse, neglect and dysfunctional households [2]. Empirical evidence has shown that these forms of early life adversity (hereafter, ELA) can have a profoundly negative impact on the individual's health outcomes in adulthood [3].

In contrast to ELA, flexibility has been identified as a protective factor that is associated with well-being in adulthood. Flexibility allows the individual to view and respond to stressful events in myriad ways, which enhances their ability to adaptively respond to adversity [4]. One

**Data Availability Statement:** The reported study was approved by the Institutional Review Board at Miami University (protocol # 01620r) and due to the sensitive nature of the data we cannot share the

data in an online data repository. Those researchers interested in replicating our findings can email me (kaliav@miamioh.edu) and/or Dr. Elizabeth Kiel (elizabeth.kiel@miamioh.edu) who serves as Chair of the Psychology Department's Review Board to seek permission from the review board committee to share de-identified data.

**Funding:** The authors received no specific funding for this work.

**Competing interests:** The authors have declared that no competing interests exist.

aspect of flexibility is cognitive flexibility. Within this manuscript, we define cognitive flexibility as the ability to switch between mental processes such as thoughts, tasks or perspectives, as described by Dajani and Uddin [5]. Cognitive flexibility then enables the individual to adjust their behavior appropriately to the environment. Within this perspective, behavioral flexibility is considered the change in behavior that occurs in response to the changes in the environment [6]. However, it is also important to note that cognitive and behavioral flexibility are frequently defined as closely intertwined concepts [6]. Cognitive flexibility is an emergent phenomenon that depends on effective executive function processes [7]. Several subdomains of executive processes (e.g., working memory, inhibition) must work in synchrony for an individual to exhibit flexible behavior [5]. For instance, working memory resources are needed to keep up with changes in environmental cues, whereas inhibition allows the individual to reduce focus on irrelevant cues. Both these processes must work competently to ensure that the individual can proceed with their goal-directed behavior. Thus, cognitive flexibility, broadly defined, characterizes one's ability to appropriately adjust their behavior to changing environmental circumstances to achieve a goal [5].

As is true of all executive processes, cognitive flexibility is implicated in activity in the prefrontal cortex (hereafter, pFC) [8, 9]. Neuroimaging studies using fMRI have shown that behavioral performance in established neuropsychological tests of cognitive flexibility are associated with increased activity in regions of the pFC [10]. The human prefrontal cortex has a protracted developmental period and is one of the last regions in the brain to achieve full maturity [11]. Additionally, the pFC is sensitive to environmental exposure, particularly experiences of stress [12].

Under normal circumstances, the amygdala, the hippocampus and the pFC form a network that allows the individual to detect threats in the environment. In response to a stressor in the environment, the body's stress response is triggered. The stress response is accompanied by the activation of the sympathetic nervous system and the hypothalamic-pituitary-adrenal (HPA) axis [13]. In the face of an immediate threat, the body's stress response is adaptive, particularly in the short-term [14]. However, for a child living in adversity, the body's normative stress response may be active much more than is helpful or adaptive as their environment may pose an ongoing threat to survival. Over time, this increases the allostatic load (i.e., biologic adaptive responses that aid in recovery from stress) and dysregulates the normative stress response. Increase in allostatic load has a negative impact on the prefrontal cortex (pFC) structure and function [14–16]. For instance, individuals who have been exposed to ELA have smaller frontal brain regions than individuals without ELA [17].

Considering that cognitive flexibility is implicated in activity in the pFC, which is negatively impacted by ELA, it is reasonable to speculate that ELA may have a deleterious effect on cognitive flexibility. Yet, the relationship between ELA and cognitive flexibility remains understudied. A few studies have examined the association between ELA and cognitive flexibility, but most of these have focused on young children [18, 19] and young adolescents [20, 21]. Findings across these studies indicate that ELA has a deleterious impact on cognitive flexibility. For instance, in a thoughtfully designed experimental study, Harms and colleagues [20] compared 14 to 17-year-old adolescents who had been exposed to ELA to a control group that had not experienced ELA. In this study, ELA was determined by physical abuse documented in local Child Protective Services records. Adolescents from the community were assigned to the non-ELA group if they had no reports of maltreatment according to both Child Protective Services records and parent responses to the Conflict Tactics Scale Parent-Child Version [22]. Finally, overall differences in exposure to ELA between the groups were confirmed using the semi-structured Youth Life Stress Interview [23]. Adolescents completed an instrumental learning task that associated rewards and punishments with picture stimuli with the goal of enhancing

rewards and reducing punishments. Adolescents with ELA performed worse than the control group in the reversal trials of this task (i.e., picture stimuli associated with rewards became associated with punishments and vice versa), which the researchers proposed was an indicator of reduced cognitive flexibility. These effects remained significant after controlling for family SES.

But, it is important to note that cognitive flexibility skills change across the lifespan [1], maturing around 10 years of age and reaching peak performance levels only between 21–30 years of age [5]. Thus, it is pertinent to examine the relationship between ELA and cognitive flexibility in adults. In contrast to the work with children and adolescents, fewer studies have examined the relations between ELA and cognitive flexibility in adults.

In one recent study, Kalia and Knauft [24] examined the association between ELA, measured using the Adverse Childhood Experiences Scale (ACEs) [25], and cognitive flexibility, assessed using the Cognitive Flexibility Inventory (hereafter, CFI) [26], in a sample of adults (N = 486). They observed that exposure to ELA was significantly and negatively associated with CFI-Control. CFI-Control is an indicator of the individual's ability to be flexible in appraising daily challenges and one of the two aspects of cognitive flexibility characterized by the CFI. CFI-Alternatives, the ability to generate alternative perspectives and solutions in the face of stress or difficulty, is the second aspect of cognitive flexibility measured by the CFI. No associations were found between CFI-Alternatives and ACEs ($r$ = -.04) [26], which may suggest that not all aspects of cognitive flexibility are associated with ACEs. Kalia and colleagues [27] also replicated these findings with a second sample of adults in a separate paper. Thus, there is some evidence to suggest that experiences of ELA, consistent with exposure to acute stress [28, 29], may be differentially associated with aspects of cognitive flexibility in adulthood with some aspects of cognitive flexibility exhibiting impairment and others being unaffected.

However, cognitive flexibility is a complex construct that exhibits both trait and state characteristics [30]. As such, it is measured using a wide variety of tasks [5, 31]. In their studies, Kalia and colleagues [24, 27] used a self-report measure that provides insight about state levels of cognitive flexibility and was intended for clinical use in response to treatment [26]. Cognitive flexibility has also been assessed in the laboratory using neuropsychological measures [31], which appear to provide insight about pFC functioning [32].

Established neuropsychological measures of cognitive flexibility, such as the Wisconsin Card Sorting Test (WCST) [33], provide an assessment of the individual's ability to switch strategies around changing rules, also known as set shifting. Measures such as the WCST engage the ability to sort information based on a particular set of rules (e.g., sort cards by color or by number) with only one rule being applicable at any given time [34]. Prior research has shown that performance on set shifting tasks is negatively impacted by acute [29] and chronic stress [35] in adults. But, the relation between performance on the WCST and ELA in adults is less studied. We present the results of two studies conducted to address this gap.

## Current study

Early developmental research on childhood adversity focused on individual risk factors, such as divorce or harsh parenting. However, more recent research has emphasized the importance of studying the cumulative effect of multiple forms of adversity, as they appear to co-occur [36, 25]. Here we use the cumulative risk model, which considers adverse early experiences in an additive manner in order to account for developmental outcomes [36], to examine the association between ELA and cognitive flexibility. The fundamental assumption of this model is that the breadth of adversity an individual experiences in early development influences outcomes.

The cumulative risk model is consistent with the theory of allostatic load, which has been used to understand the impact of ELA on pFC structure and function.

We examined the association between ELA and cognitive flexibility in two separate samples: college students and adults. Based on prior research by Kalia and colleagues [24, 27], our primary hypothesis was that increased exposure to ELA, assessed using the Adverse Childhood Experiences Scale (ACEs) [25], would be negatively associated with cognitive flexibility measured using the Wisconsin Card Sorting Test [33]. Since ELA dysregulates the normative stress response system resulting in an increased allostatic load [14], our second hypothesis was that the number of ACEs reported would correlate positively with perceived chronic stress levels. Additionally, as chronic stress is a well-known risk factor for negative mental and physical health outcomes [37] and impairs pFC function [38], we also explored the relationship between perceived chronic stress and cognitive flexibility. As prior research has shown that perceived chronic stress can reduce cognitive flexibility in college students [35], our third hypothesis was that we would observe a significant negative association between perceived chronic stress and performance on the WCST. Since cognitive flexibility skills develop slowly and appear to decline with aging [39], our final hypothesis was that age would be negatively correlated with performance on the WCST within the adult sample.

## Study 1 methods

### Participants and procedure

Undergraduate students (N = 215; Women = 139; $M_{age}$ = 19.1) were recruited from a Midwestern university subject pool to participate in exchange for course credit. Participants, who were undergraduates currently enrolled in the university, were included in the study if they came to the laboratory and signed a consent form agreeing to participate. No other inclusion or exclusion criteria were used. Most participants (73.0%) identified as White. The remaining participants identified as Asian (15.8%), African American or Black (3.2%), Biracial (2.3%), Hispanic (2.3%), Native American (0.9%) or a race or ethnicity not listed (2.3%). Data collection took place in the first author's laboratory. Following informed consent, participants completed questionnaires and cognitive tasks as part of a larger study. Total participation lasted approximately 45 minutes. All study procedures were approved by the Miami University Institutional Review Board (Protocol # 01620r). All participants provided written informed consent prior to completion of the following measures and demographics.

### Measures

**Adverse childhood experiences scale (ACEs) [25].** The ACEs Scale is a scale that assesses exposure to childhood abuse and household dysfunction across 17 items and seven categories of abuse and dysfunction. Participants are asked to rate if they had experienced or not experienced that aspect of abuse or household dysfunction prior to the age of 18. There are three categories that assess types of abuse: psychological, physical, and sexual, which include questions such as "*Did a parent or other adult in the household often or very often hit you so hard that you had marks or were injured*". The remaining four categories assess household dysfunction: substance abuse, mental illness, mother being treated violently, and having an incarcerated household member, which included questions such as "*Was a household member depressed or mentally ill*?". All categories except having an incarcerated household member are assessed by multiple items. Consistent with Felitti and colleagues [25], every time participants responded affirmatively ("yes") to at least one question in a category, they were given one point for that category. Consistent with Felitti and colleagues [25], we then summed the number of categories reported, which acted as the total ACEs score. Thus, an individual who responded

affirmatively to both questions in the physical abuse category, one question in the psychological abuse category, and one question in the substance abuse category would have an ACEs score of three. Consequently, higher ACE scores indicate greater exposure to early adversity.

**Perceived stress scale (PSS) [40].** Emerging from the transactional model of stress [41], the PSS measures appraisals of stress over the past month, rather than the counting of "objectively" stressful events [40]. Across this and other studies, Cohen and colleagues [41–43] have found perceived stress to predict health outcomes and behaviors. The PSS is a ten-item questionnaire assessing how stressful, overwhelming or out of their control participants perceived their lives over the past month to have been (e.g., *In the last month, how often have you found that you could not cope with all the things you had to do*?). Each item is rated on a five-point Likert scale ranging from "*never*" (1) to "*often*" (5). Scores are reverse coded as needed and summed to create a total perceived stress score. Higher scores indicate more perceived stress (α = .85).

**Wisconsin card sorting test (WCST) [33].** A computerized version of the WCST was used to measure cognitive flexibility. The WCST is an instrument developed by Grant and Berg [44] to measure flexibility in thinking through sorting cards along multiple dimensions. The WCST measures cognitive flexibility by asking participants to adapt their card sorting behavior to changing rules, which requires them to continually shift their response strategies [45, 46]. Within the WCST, participants are shown a pile of cards to be sorted one at a time. Each of these cards can be sorted into one of four piles, depending on what dimension of the card participants choose to sort by. In essence, there are three potential sorting rules, and participants must guess which is the correct sorting rule through trial and error. After each card is sorted, participants received feedback of either "Correct" or "Incorrect" notifying them whether or not they had sorted by the appropriate rule [47]. Once participants sort six consecutive cards by the correct rule, the sorting rule changes, and participants must adjust their strategy to find the new rule. The task consisted of 48 trials.

The WCST provides a range of performance measures. The outcomes of interest for the present study are total correct, perseverative errors and completed categories.

*Total Correct.* Total correct is the total of trials in which the participants sorted the cards correctly [44, 48].

*Perseverative Errors.* Perseverative errors indicate the number of times participants persisted in sorting cards by a previously correct, but no longer accurate rule, despite receiving feedback that they had made an error in sorting [44, 48].

*Completed Categories.* Completed categories provide insight about the number of sorting rules (i.e., sort by numbers, colors, or shapes) that the participant successfully identified and learned correctly. Each sorting rule could be presented twice, meaning participants could successfully complete up to six categories. Categories are considered "completed" when the participant makes 6 correct trials in a row within a sorting category [48, 49].

Total correct, perseverative errors and completed categories have been found to load together on the first factor in factor analyses of the WCST [50], which is thought to reflect shifting and flexibility [51]. Of these, perseverative errors is the outcome most frequently reported to measure cognitive flexibility [34, 45, 47, 52].

## Study 2 methods

### Participants and procedure

Data were collected over two days via the online platform Prolific in exchange for $4.50. Eligible participants were adults 18 years and older residing in the United States. Monte Carlo simulations anticipating small to medium effect sizes were used to determine the number of

participants recruited. The final sample size (N = 295) emerged after 3 participants failed more than 1 attention check. The final sample consisted of Men = 160, Women = 133, Non-binary = 1, Prefer not to describe = 1. Participant ages ranged from 18–80 years ($M_{Age}$ = 36.24, 94.6% were younger than 61 years). The majority (93.9%) self-identified as middle-class on a self-anchoring scale in the form of a 10-rung ladder; 65.4% reported having either a bachelor's or associate's degree or having completed some college. Most participants identified as White (80.0%), and the remaining participants self-categorized as African American or Black (6.8%), Asian or Asian American (5.4%), Hispanic or Latino/a/x (5.8%), Indian American (.7%), Native American (.3%), a race or ethnicity not listed (0.7%; both self-identified as mixed-race) or prefer not to disclose (0.3%). All study procedures were approved by the Miami University Institutional Review Board. All participants provided written informed consent prior to participation in the study and then completed the following measures in addition to demographics.

## Measures

**Adverse childhood experiences (ACEs) [22].** Instead of the 17-item ACEs measure used by Felitti and colleagues [25], we used the 10-item ACEs [53]. This was done to reduce the burden on participants as the data were collected online.

The 10 item ACEs measure used in Study 2 is similar to that which was used in Study 1, with two primary differences. First, multiple questions addressing each category of ACEs have been combined into a single item, as used by Stein and colleagues [53]. For example, the two questions for psychological abuse ("*Did a parent or other adult in the household often swear at you insult you, put you down, or humiliate you?*" and "*Did a parent or other adult in the household often act in a way that made you afraid you might be physically hurt?*") were combined into the single item "*Did a parent or other adult in the household often swear at you, insult you, put you down, or humiliate you OR act in a way that made you afraid you might be physically hurt?*". Second, 1 item assessing emotional neglect *("Did you often feel that your family didn't look out for each other, feel close to each other, or support each other")* and 1 item assessing physical neglect ("*Did you often feel that you didn't have enough to eat, had to wear dirty clothes, and had no one to protect you?*") were added. This ACEs scale consists of 10 items assessing an individual's exposure to early life adversity. Five items in the scale ask about exposure to different types of maltreatment (e.g., *Did a parent or other adult in the household often push, grab, slap or throw something at you?*), and five items request information about parental or family incapacities (e.g., *Were your parents ever separated or divorced?*). Every response in the affirmative ('yes') to a question was given 1 point. To account for the cumulative effect of adverse childhood experiences, we summed each individual's "yes" responses to calculate their ACE score. Consequently, higher ACE scores indicate broader exposure to adverse experiences in early development.

**Perceived stress scale (PSS) [40].** As in Study 1, the PSS was used to measure perceived stress. Higher PSS score indicated more perceived stress (α = .91).

**Wisconsin card sorting test (WCST) [33].** Like Study 1, we used the computerized version of WCST to measure cognitive flexibility. The WCST was administered online through Inquisit Web. Through this web-based program, we were able to direct participants to a link which allowed them to complete the WCST at the end of our survey. We focused once again on measures of total correct, perseverative errors and completed categories as our outcomes of interest.

## Data processing and analytic plan

In order to interrogate the relationships between ACES total, PSS, total correct, perseverative errors, completed categories and age, bivariate correlations were conducted separately for the

two studies. Assuming that ACEs correlate with performance on the WCST, hierarchical regression analysis was carried out to determine whether ELA is uniquely associated with cognitive flexibility. Since there is some empirical evidence that performance on the WCST is influenced by gender and age [54], these variables were added as control variables in the regression equations for both the college students and adults. Additionally, as prevalence of ACEs differs by race/ethnicity [55], this variable was also added as a control variable to the regression equations for both college student and adult samples. College students tend to be more homogenous than adult samples [56]. Therefore, both education, which influences performance on the WCST [39, 54] and SES, which is associated with the prevalence rate of ACEs [57], were included as control variables in the regression analyses for the adult sample.

## Results

Table 1 presents the descriptive statistics of all relevant variables. Examination of the distribution of the data showed that the skewness and kurtosis of number of ACEs, total correct score on the WCST and number of perseverative errors on the WCST were greater than |2| for Study 1. Similarly, number of perseverative errors and total correct scores had skewness and kurtosis greater than |2| for Study 2. Prior to conducting any further analyses, we used the Blom's rank-based inverse normal transformation only on the variables identified above, as described in Bishara and Hittner [58]. Following the transformation, the data were checked again and were observed to be normally distributed.

### Bivariate correlations

Bivariate correlations, to examine the association between ACEs and cognitive flexibility, were conducted separately for Study 1 and Study 2 (see Table 2). The findings revealed that the number of ACEs reported was negatively correlated with cognitive flexibility in both the samples.

**Table 1. Demographic characteristics and descriptive statistics.**

| | Study 1 (College aged adults) | Study 2 (Adults) |
|---|---|---|
| | Prolific (N = 215) | Prolific (N = 295) |
| Characteristics | N (%) | N (%) |
| *Gender* | | |
| Man | 75 (35%) | 160 (54.2%) |
| Woman | 139 (64.7%) | 133 (45.1%) |
| Non-binary | - | 1 (.3%) |
| *Race* | | |
| White | 157 (73%) | 236 (80%) |
| African American | 7 (3.2%) | 20 (6.8%) |
| Native American/Alaskan Native | 2 (.9%) | 1 (.3%) |
| Asian or Asian American | 34 (15.8%) | 16 (5.4%) |
| Hispanic or Latino | 5 (2.3%) | 17 (5.8%) |
| Other | 10 (4.6%) | 5 (1.7%) |
| *ACEs* | | |
| 0 | 132 (61.4%) | 101 (34.2%) |
| 1 | 48 (22.3%) | 55 (18.6%) |
| 2 | 17 (7.9%) | 38 (12.9%) |
| $\geq$ 3 | 17 (7.9%) | 101 (34.4%) |

**Table 2. Descriptive statistics and bivariate correlations between relevant variables.**

| Study | | Mean | SD | 1 | 2 | 3 | 4 | 5 | 6 |
|---|---|---|---|---|---|---|---|---|---|
| 1 | 1. ACEs total | .66 | 1.08 | - | .16* | -.11 | -.13 | -.17* | .10 |
| | 2. PSS | 27.87 | 6.02 | | - | .01 | -.09 | .02 | -.01 |
| | 3. Total correct | 32.91 | 6.63 | | | - | .06 | .76*** | .01 |
| | 4. Perseverative errors | 4.51 | 2.21 | | | | - | .17* | .01 |
| | 5. Completed categories | 4.79 | 1.45 | | | | | - | -.08 |
| | 6. Age (years) | 19.10 | 1.42 | | | | | | - |
| 2 | 1. ACEs total | 2.19 | 2.50 | - | .24*** | -.05 | -.12* | -.16** | .04 |
| | 2. PSS | 26.96 | 7.48 | | - | .03 | .01 | -.04 | -.20** |
| | 3. Total correct | 69.90 | 12.72 | | | - | .32*** | .46*** | -.08 |
| | 4. Perseverative errors | 7.66 | 4.35 | | | | - | 44*** | .08 |
| | 5. Completed categories | 4.86 | 1.80 | | | | | - | -.13* |
| | 6. Age (years) | 36.23 | 12.14 | | | | | | - |

Note.

* $p < .05$;

** $p < .01$;

*** $p < .001$.

In Study 1, the number of ACEs was negatively correlated with the number of completed categories (see Table 2) and marginally associated with perseverative errors ($r = -.13$, $p = .055$) on the WCST. Additionally, the number of reported ACEs was positively correlated with perceived chronic stress ($r = .16$, $p = .02$). No other correlations emerged as significant. Perceived chronic stress was not significantly correlated with any of the measures of cognitive flexibility. See Table 2.

In Study 2, the number of ACEs was negatively correlated with the number of completed categories ($r = -.16$, $p = .005$) and perseverative errors ($r = -.12$, $p = .033$). Additionally, reported number of ACEs was positively correlated with perceived chronic stress ($r = .24$, $p = .001$). Finally, ACEs was negatively correlated with education levels ($r = -.17$, $p = .003$), and age was negatively correlated with the number of categories completed on the WCST ($r = -.13$, $p = .02$). Perceived chronic stress was not significantly correlated with performance on the WCST. See Table 2.

### Predicting cognitive flexibility: Study 1

To assess the unique association between ACEs and cognitive flexibility, we conducted one hierarchical regression analysis. In the regression model, ACEs was the primary predictor variable and number of completed categories was the outcome variable. Participants' age, gender and race/ethnicity were added as control variables in the first step of the regression equation. See Table 3.

The model predicting the number of completed categories was significant, accounting for 4.8% of the variance in the score, $F (4, 204) = 2.56$, $p = .04$. Change in $R^2$ between model 1 and model 2 (with ACEs as a predictor) $= .03$, $F_{change}(1, 204) = 6.45$, $p_{change} = .012$. The Durbin Watson test statistic $= 2.0$, indicating that there was no evidence of autocorrelation amongst the residuals [59]. In support of our primary prediction, ACEs ($t = -2.55$, $p = .012$; 95% CI [-.57, -.07]) emerged as a significant predictor. Individuals who reported higher numbers of ACEs also demonstrated fewer completed categories on the WCST. No other predictor emerged as significant.

**Table 3. Hierarchical multiple regression predicting completed categories on WCST in Study 1.**

| N = 215 | | Completed categories | | |
| --- | --- | --- | --- | --- |
| | Independent variables | $R^2$ = .05* | | |
| Step | | B | SE | β |
| 1 | Age | -.07 | .07 | -.07 |
| | Gender | .03 | .22 | .01 |
| | Race/ethnicity | .07 | .05 | .10 |
| 2 | ACEs total | -.32 | .13 | -.18* |

Note.

* $p < .05$;

** $p < .01$;

*** $p < .001$.

## Predicting cognitive flexibility: Study 2

To assess the unique association between ACEs and cognitive flexibility, we conducted two hierarchical regression analyses. Participants' age, gender, race/ethnicity, education and SES levels were added as control variables in the first step of the regression equations. In the second step, participants' reported number of ACEs was added as the primary predictor variable into the regression equations.

The overall model for perseverative errors was not significant, $F$ (6, 288) = 1.29, $p$ = .26. See Table 4. However, the model predicting the number of completed categories was significant, accounting for 6.7% of the variance in the score, $F$ (6, 288) = 3.44, $p$ = .003. Change in $R^2$ between model 1 and model 2 (with ACEs as a predictor) = .024, $F_{change}$(1, 288) = 7.47, $p_{change}$ = .007. The Durbin Watson test statistic = 2.11, indicating that there was no evidence of auto-correlation amongst the residuals [59]. In support of our primary prediction, ACEs ($t$ = -2.73, $p$ = .007; 95% CI [-.20, -.03]) emerged as a significant predictor, such that individuals who reported higher number of ACEs also demonstrated fewer completed categories on the WCST. Age also emerged as significant predictor of categories completed ($t$ = -2.78, $p$ = .006;

**Table 4. Hierarchical regression predicting performance on WCST in Study 2.**

| N = 295 | | Perseverative error | | | Completed categories | | |
| --- | --- | --- | --- | --- | --- | --- | --- |
| | Independent variables | $R^2$ = .03 | | | $R^2$ = .07** | | |
| Step | | B | SE | β | B | SE | β |
| 1 | Age | .01 | .01 | .09 | -.02 | .01 | -.17** |
| | Gender | .02 | .11 | .01 | .19 | .20 | .06 |
| | Race/ethnicity | -.03 | .04 | -.05 | .08 | .07 | .07 |
| | Education | .03 | .04 | .04 | .16 | .08 | .14* |
| | SES | -.01 | .04 | -.02 | -.01 | .07 | -.01 |
| 2 | ACEs total | -.05 | .02 | -.13* | -.12 | .04 | -.16** |

Note.

* $p < .05$;

** $p < .01$;

*** $p < .001$.

95% CI [-.04, -.007]), such that older individuals reported lower scores on number of completed categories.

## Discussion

In the most recent survey by Giano and colleagues [60] of the prevalence of ELA in the United States, 57.8% of participants reported experiencing at least 1 ACE and 21.5% reported experiencing 3 or more ACEs [60]. Data from our work is consistent with these numbers. We observed that 38.6% of the college students in Study 1 and 65.5% of the community sample in Study 2 reported experiencing ACEs. Our data reveal that the college students reported fewer ACEs, on average, than the adult sample ($t$ = 8.42; $p$ = .001). Since there is empirical evidence showing that ACEs increase the risk of lower educational outcomes, it is possible to speculate that individuals with higher ACEs opted out of a college education [61]. This may be part of the reason why our sample of college students consists of individuals with fewer ACEs in comparison to the community sample. Regardless, this finding suggests that experience with ELA is, unfortunately, commonplace for the average American. Considering that ACEs are associated with deleterious physical and mental health outcomes [62, 63], it is imperative that the relationship between ELA and protective factors associated with enhanced mental and physical health are investigated.

Using the cumulative risk model [36], we examined the relationship between ACEs and cognitive flexibility, which is a factor associated with improved health and wellbeing in adulthood [4, 24]. Our primary prediction was that the number of ACEs would be negatively correlated with performance on the cognitive flexibility task, namely the WCST. In both studies, we observed that the number of ACEs reported by college students and adults was significantly and negatively correlated with the number of categories completed on the WCST. To the best of our knowledge, we are the first to report this association in adults. Further, hierarchical regression analyses indicated that the number of ACEs was uniquely associated with lower scores on categories completed after controlling for relevant variables (e.g., gender, age) in both college students and adults. Thus, the data provided support for our hypothesis. These findings are consistent with prior reports demonstrating that ACEs are associated with reduced cognitive flexibility in adolescents [20] and adults [24, 27].

An individual's score of the number of completed categories is an indicator of their overall performance on the WCST [64]. In order to complete a category, participants had to learn the relevant sorting rule (e.g., sort cards by numbers) and apply it correctly for 6 trials until the rule changed. Consequently, the score for completed categories provides evidence that the individual is able learn specific stimulus response contingencies (e.g., the card in hand is sorted with the card that has the same number) and adaptively adjust these learned contingencies based on changing environmental demands (e.g., sorting rule changes from number to color). The negative association between number of ACEs and number of completed categories suggests that breadth of exposure to ELA impairs an individual's ability to learn and adaptively adjust stimulus response contingencies. This is consistent with findings reported by Harms and colleagues [20] with adolescents who had experienced ELA.

Interestingly, we observed that the correlation coefficients between ACEs and the number of categories completed did not differ between the college students and adult sample (Fisher's z = -.11; $p$ = .45). This suggests that it is possible that ELA alters fundamental processes in attention and associative learning [20]. However, examination of the violin graphs (see Fig 1) indicates that the distribution of the data in the college student sample is not the same as the adult sample. The violin graphs represent the distribution of the data around the median, which is less sensitive than the mean to extreme scores [65]. As the black line in the graphs

(a)

(b)

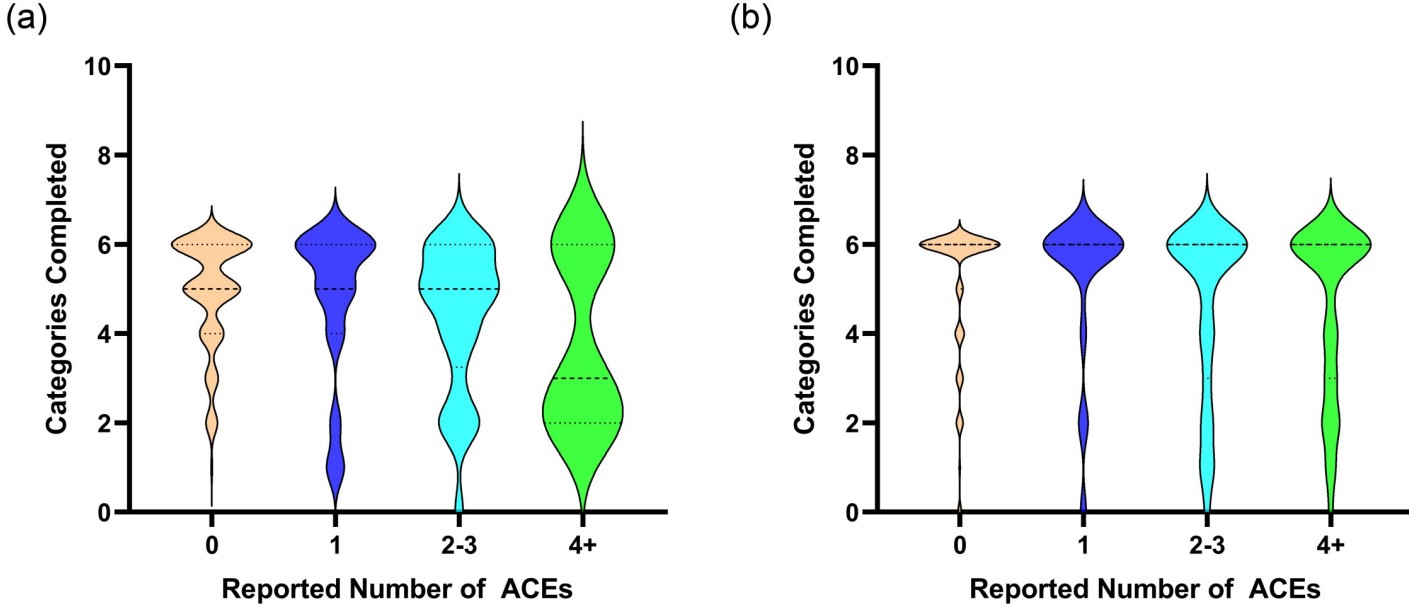

**Fig 1. Distributions for the college sample (a) and online adult sample (b) of completed categories for individuals reporting 0, 1, 2–3, and 4+ ACEs before the age of 18.** The center of each distribution is identified by the median, which can be identified by the thicker line with larger dashes. The 25th and 75th quartiles can be identified by the thinner lines with smaller dashes.

indicates, median values for the college students were lower than median values for adults. Additionally, the median number of completed categories for individuals who had reported 4 or more ACEs was lower than those who had fewer ACEs, but only in the college student sample. This may suggest that the relationship between ACEs and cognitive flexibility could differ in college students versus community adults. It is important to note that a small difference in neuropsychological measures may not translate into behavioral differences in the real world. So, our study cannot directly address the impact, if any, this may have in an individual's life. But, longitudinal research that examines the association between ELA and cognitive flexibility across the lifespan will be able to provide further insight on this issue.

It is relevant to point out that previous research on ACEs and cognitive flexibility in adults have primarily examined state characteristics of cognitive flexibility using self-report measures. Specifically, those reports had observed a negative association between ACEs and an individual's flexibility in appraising everyday challenges as controllable [24, 27]. A few studies have investigated associations between ACEs and cognitive flexibility using behavioral measures of shifting by focusing on adults with depression and bipolar disorder [66] and mothers with children 5 years and younger [67]. These have also observed a negative association between ELA and cognitive flexibility. In contrast, Mittal and colleagues [68] found that ELA was associated with enhanced shifting in adults. It is important to note that the work done by Mittal and colleagues differed from ours in two important ways. First, they focused only material and financial uncertainty of the childhood environment as a source of ELA, arguing that cognitive flexibility may be an adaptive response to unpredictable childhood environments. Second, they assessed cognitive flexibility using the color-shape task rather than the WCST. Considering the limited number of studies that have examined the association between ELA and cognitive flexibility using the WCST, our work adds to the existing literature.

According to the model outlined by Schaie [69], inflexibility can be structural, attitudinal and functional. Inflexible appraisal of everyday challenges would be an example of attitudinal flexibility, whereas inflexible categorization of cards in a sorting task would qualify as an example of functional inflexibility. Thus, our data suggest that individuals exposed to ELA exhibit reduced functional flexibility in adulthood. One real-life implication of this finding is that the adult with ELA may struggle to be flexible in the deployment of strategies in their pursuit of goals. As detailed below, perseveration was not uniquely associated with ACEs in either college students or community adults. Thus, it is possible to speculate that individuals with ELA may be less likely to perseverate with a single strategy. Instead, our data suggest they may have trouble figuring out which strategy is appropriate for the moment. Consequently, social support or the opportunity to 'talk things through' with someone may be an important predictor of resilience in adults with ELA [70]. Future research should consider examining the relationship between ACEs and flexibility across several domains, including interpersonal relationships.

In addition to the number of completed categories, ACEs was marginally negatively correlated with perseverative errors in college students and significantly negatively correlated with perseverative errors in adults ($r$s = -.13 and -.12 respectively). The marginally significant correlation between ACEs and perseverative errors in college students may be due to the fact that cognitive flexibility continues to develop into the third decade of an individual's life [5]. Interestingly, the negative correlation in adults suggests that breadth of experience with ELA is associated with less rigidity. Perseveration on the WCST is the most frequently reported measure of cognitive flexibility and an indicator that the person ignored feedback to persist with a dominant response that was no longer applicable. The negative association between ACEs and perseverative errors suggests that adults with ACEs are able to switch between strategies in goal pursuit. However, it is relevant to note that after accounting for the covariates the overall model predicting perseverative errors did not emerge as significant. Regardless, our work should provide impetus for future research on the relationship between ELA and perseveration.

Finally, participants' total correct scores on the WCST were not associated with ACEs in either college students or community adults. Prior research by Kalia and Knauft [24] has shown that exposure to ACEs was not associated with an individual's ability to take alternative perspectives and generate alternate solutions to problems, as measured by the *Alternatives* subscale of the CFI [24]. This finding is consistent with the observation that not all aspects of cognitive flexibility are negatively impacted by exposure to ELA. Some researchers have suggested that experiencing ELA may provide individuals with some cognitive benefits [68, 71], including improved detection of dangers. Considering the limited amount of research on this topic, future research should consider examining associations between ACEs and cognitive benefits.

Contrary to our hypothesis, perceived chronic stress was not associated with any of the measures of cognitive flexibility assessed by the WCST. Even though there have been prior instances where an association between perceived chronic stress and cognitive flexibility has been observed [24], there have also been reports where no association between the two variables was reported [28]. In a thoughtfully designed experimental study, Liston and colleagues [35] had observed that perceived chronic stress was associated with impaired attention shifts in an fMRI compatible visual discrimination task. Since cognitive flexibility is a complex construct that depends on effective executive functions [5], it is possible that perceived chronic stress influences some aspects of executive processes and not others. More research needs to be done on the relationship between perceived chronic stress and cognitive flexibility before any clarity can be achieved. However, we did observe that ACEs positively correlated with perceived chronic stress in both samples. Thus, our data indicate that individuals with ACEs are more likely to be experiencing higher levels of chronic stress. Considering that perceived

chronic stress is associated with changes in pFC functioning [35] and has emerged as a key mediator between ACEs and flexibility in other work [24], future research should examine perceived chronic stress in relation to ACEs and executive processes.

Finally, age emerged as an independent predictor of reduced cognitive flexibility in the community sample. This finding is consistent with prior research indicating that cognitive flexibility declines with aging [39]. Recent research has shown that experiencing 3 or more ACEs increases the chances of developing dementia after accounting for age, sex, education, childhood economic hardship and nutrition [72]. Although we did not observe significant interaction effects between ACEs and age on cognitive flexibility in our adult sample, future research should examine the role of these two variables on cognitive flexibility in aging adults.

The findings of our study should be interpreted with a view toward the limitations. First, our findings are novel and need to be replicated before any firm conclusions can be drawn. Additionally, it is relevant to note that we used the 17-item ACEs scale for Study 1 and the 10-item scale for Study 2. This limits our ability to truly compare findings across the two studies presented here. Second, our data are cross-sectional in nature, which precludes us from making any causal claims. Additionally, the variance explained by our regression analyses was rather small, which suggests there is individual-level variability that may be influencing the association between ACEs and cognitive flexibility. Fourth, we did not collect data on the current economic status of the college students. As such, we cannot determine the role that economic hardship may have played in their cognitive flexibility or perceived chronic stress levels. Although the observed relationships between ACEs and cognitive flexibility were consistent between the samples and SES did not emerge as a predictor in the adult sample, presence or absence of economic hardship may have influenced the data in the college sample. Additionally, we do not have information about childhood economic hardship for either the adults or the college students. This prevents us from knowing whether childhood poverty also influenced the development of cognitive flexibility. Finally, we do not know how many adults in the community sample that reported having 'some college' experience were enrolled in college courses at the time of data collection. Nevertheless, our study provides further evidence that experience with ACEs is associated with reduced cognitive flexibility. Further, by examining cognitive flexibility using the WCST, we were able to extend the limited literature on this topic.

## Author Contributions

**Conceptualization:** Vrinda Kalia.

**Data curation:** Vrinda Kalia.

**Formal analysis:** Niki Hayatbini.

**Methodology:** Vrinda Kalia.

**Project administration:** Vrinda Kalia.

**Resources:** Vrinda Kalia.

**Supervision:** Vrinda Kalia.

**Visualization:** Katherine Knauft.

**Writing – original draft:** Vrinda Kalia.

**Writing – review & editing:** Katherine Knauft, Niki Hayatbini.

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
