## [Decision Letter · Decision Letter 0]

9 Aug 2021

PONE-D-21-09458

Adverse childhood experiences (ACEs) associated with reduced cognitive flexibility in both college students and adults

PLOS ONE

Dear Dr. Kalia,

Thank you for submitting your manuscript to PLOS ONE. After careful consideration, we feel that it has merit but does not fully meet PLOS ONE’s publication criteria as it currently stands. Therefore, we invite you to submit a revised version of the manuscript that addresses the points raised during the review process.

We look forward to receiving your revised manuscript.

Kind regards,

Torsten Klengel, MD PhD

Academic Editor

PLOS ONE

Journal Requirements:

2. Please change "female” or "male" to "woman” or "man" as appropriate, when used as a noun (see for instance https://apastyle.apa.org/style-grammar-guidelines/bias-free-language/gender).

Reviewers' comments:

Reviewer's Responses to Questions

**Comments to the Author**

1. Is the manuscript technically sound, and do the data support the conclusions?

Reviewer #1: Yes

Reviewer #2: Yes

2. Has the statistical analysis been performed appropriately and rigorously? 

Reviewer #1: Yes

Reviewer #2: Yes

3. Have the authors made all data underlying the findings in their manuscript fully available?

Reviewer #1: Yes

Reviewer #2: Yes

4. Is the manuscript presented in an intelligible fashion and written in standard English?

Reviewer #1: Yes

Reviewer #2: Yes

5. Review Comments to the Author

Reviewer #1: Abstract

1. Last sentence reads reduced flexibility in adulthood. But, the results only focus on reduced cognitive flexibility.

Intro

1. The authors should consider introducing ACEs in a slightly different way to encompass what ACEs are more fully. Instead of saying: “…humans who do not get the parental attention they need in early development” they could say: “humans who do not get basic needs met in early development”, as ACEs can consist of many things aside from concerns pertaining to parental attention

2. For the paragraph pertaining to flexibility: the author could benefit from briefly defining both cognitive and behavioral flexibility before providing the elaboration on how cognitive flexibility impacts behavioral flexibility.

3. For the Harms study, what was ELA measured by? How much words did the ELA group perform than controls? Was it meaningful?

4. In the Kalia and Knauft study, participants only performed poorly on the CFI-control. What is the other aspect assessed on the CFI? Why might participants have performed “fine” on that?

Method

1. Please clarify inclusion and exclusion criteria for each sample

2. Please clarify if participants in their second sample (adults) excluded adults currently attending college

3. When describing the ACEs, the author may want to align their categories more closely to what the original developer (Feletti) identified them as. Specifically, “criminal behavior in the household” should be changed to “incarcerated household member”, as these insinuate two fairly different experiences

Analysis

1. Language should be changed to how the analysis was conducted, not their plan for the analysis (written in future tense)

2. Author states that they used Blom’s rank-based inverse normal transformation on the relevant variables; authors should specific which ones these variables are, or if this was used on all variables

Results

Discussion

1. First sentence describes rates for at least 1 ACE and then 3 or more, but college sample is described as 38.6% experiencing ACE’s. Why the difference in reporting?

2. Author could benefit from looking into “post traumatic growth” literature to check if flexibility in relation to ELA is common; mentioned that there is little research on this subject, but may be more within the subfield of positive psychology

3. Perseverative errors is the outcome most frequently reported to measure cognitive flexibility – but this was only significant in adult sample. Should address this more in Discussion.

4. What about total correct not being significant – why not addressed in discussion?

5. Address that variance was rather small for regressions.

6. How meaningful are these results – meaning how far apart were scores on WCST? A significant difference in neuropsych results does not always equate to meaningful functional differences.

7. Limitations – note differences between 17 item ACES and 10 item ACES

8. Spell out how ELA could provide some cognitive benefits.

Reviewer #2: Minor grammatical errors throughout (e.g., inappropriate use of semi-colons, run on sentences, etc.). For example, the semi-colons on line 27 and on line 78 should be replaced with commas. These grammatical errors are minor and a final review/polish of the manuscript is recommended.

The methods of data analysis and regression models are well done and nicely presented.

It would be interesting to have a bit more discussion on the implications of your findings and what it means for adults who've experienced adverse events as children. For example, how would this reduced ability to shift-focus affect people in their jobs, education, or at home/taking care of a family?

Nice paper that demonstrates a significant finding; as a suggestion for future research, it would be interesting to repeat this in a population that had experienced greater amounts of trauma.

6. PLOS authors have the option to publish the peer review history of their article (what does this mean?). If published, this will include your full peer review and any attached files.

Reviewer #1: No

Reviewer #2: No

---

## [Author Response · Author response to Decision Letter 0]

7 Sep 2021

Please see the letter attached as response to reviewers.

---

## [Editor Report · Decision Letter 1]

18 Nov 2021

Adverse childhood experiences (ACEs) associated with reduced cognitive flexibility in both college and community samples

PONE-D-21-09458R1

Dear Dr. Kalia,

We’re pleased to inform you that your manuscript has been judged scientifically suitable for publication and will be formally accepted for publication once it meets all outstanding technical requirements.

Kind regards,

Torsten Klengel, MD PhD

Academic Editor

PLOS ONE
---

## [Editor Report · Acceptance letter]

19 Nov 2021

PONE-D-21-09458R1 

Adverse childhood experiences (ACEs) associated with reduced cognitive flexibility in both college and community samples 

Dear Dr. Kalia:

I'm pleased to inform you that your manuscript has been deemed suitable for publication in PLOS ONE. Congratulations! Your manuscript is now with our production department. 

Kind regards, 

on behalf of

Dr. Torsten Klengel 

Academic Editor

PLOS ONE